# Women Academics in the World of Neoliberal, Managerial Higher Education

**Margaret Sims**

Department of Educational Studies, Faculty of Human Sciences, Macquarie University, Sydney, NSW 2109, Australia; margaret.sims@mq.edu.au

**Abstract:** In my last years in academia, I have experienced the intimidating impact of pettybureaucracy and top-down micromanagement that typify managerialism in higher education today. In this paper I use my own experiences to reflect on why this is happening, attempting to gain understanding that can support others still working in the sector to survive and ultimately thrive. I argue that neoliberalism operates as an ideology, shaping the way we perceive and act in the world. In higher education, it is enacted through managerialism, together creating a social imaginary that defines what is expected of managers and what is expected of workers. Women are particularly vulnerable in this social imaginary given that the challenges they face in the workforce are attributed to their own shortcomings rather than any systemic barriers. Women face choices as to how to operate in this social imaginary, but all choices have consequences that need to be understood and managed. Ultimately, systemic disadvantage will not change without significant action taken by collectives of women who have a clear vision of better alternatives.

**Keywords:** higher education; gender; neoliberalism; managerialism





## 1. Introduction

In my last years in academia, I have experienced "the petty self-perpetuating creation of needless bureaucracy and anti-professional controls" [1] (p. 5) that typify managerialism in higher education today. From working in a job that I once loved, I gradually became the target of a relentless parade of petty acts of disrespect, positioned as untrustworthy, unprofessional and a trouble-maker. The sector I joined many years ago with stars in my eyes no longer exists and has been replaced by a corporate neoliberal entity that treats workers as cattle fodder—human capital to be manipulated and discarded at a whim. Unfortunately, women are particularly disadvantaged in this (not) brave new world given they are more likely to be employed at the lower levels within the university (both professional and academic) and are less likely to be promoted. In this paper I use my own experiences along with the literature to create a framework that enables me to reflect on why this is happening, attempting to gain understanding that can support others still working in the sector to survive and ultimately thrive.

## 2. Neoliberalism

In doing this I use the framework proposed in my book [2]. This framework is generic in that it addresses the impact of neoliberalism and managerialism in higher education; however, in this paper I apply the framework specifically to address the experiences of women academics. The framework identifies neoliberalism as the overarching ideology that shapes the way that we look at the world and understand what is happening to us, and how it crafts our behavioural responses to our experiences [2]. Neoliberalism thus creates a context where the various elements combine to create a social imaginary [3] or zeitgeist. As part of this process, certain elements and understandings are identified as important, of more value than others [4], and conversely, other elements are rendered unimportant

and of little or no value. This social imaginary is responsible for the creation of wider and wider gaps between those who are privileged and those who are not with inequality currently greater now than any time in the past 100 years [5–10]. As explained by Bettache and Chiu [11] (p. 217), neoliberalism condones:

> *social inequality by attributing the presence of social hierarchy to innate individual differences (e.g., IQ) and acquired traits . . . This attribution style also serves to legitimize the suffering of structurally disadvantaged groups, such as minorities or people with mental health problems.*

In this article, I position women academics as one of the disadvantaged groups identified by Bettache and Chiu above.

## 3. Management and Managerialism

Given the existence of privilege, it is not surprising that members of a society seek to join the elite. In the higher education sector, for many the route to achieve this is through management. Increasingly management is perceived as a set of skills that ought to be applied independently of the sector in which managers are working so that managers in higher education are increasingly appointed from positions outside the sector [12]. In other words, management skills are generic, and managers ought to remain as separate from those they manage as possible.

Social representation theory [13] can be used explore the experiences of those working in management roles. The theory proposes that humans seek to belong to groups of people who understand the world in ways similar to themselves. Once membership is established in such groups, this shared understanding influences how group members perceive new experiences, as well as their behavioural responses to those experiences. Applying this idea to the higher education sector and the proliferation of managerialism therein, it can be argued acceptance in the management group not only requires demonstration that the "management" way of seeing the world is shared, it also requires demonstration of loyalty to the group through management-defined "acceptable" behaviours. In this context, the neoliberal social imaginary has shaped our understanding of management and what is required to be a good manager. In particular, success in management requires demonstration of masculinist leadership behaviours. These behaviours, used to demonstrate one's belonging to the management group, are identified as "aggressive, competitive, self-centred and emotionally cold" [14] (p. 127), in other words, successful demonstration of belonging required one to "manage like a man" (ibid). For those seeking to join the elite through promotion into the professoriate, it is my experience it is increasingly common that the successful path is policed by management whose definitions of success influence what is considered acceptable behaviour. Those who are not able to demonstrate outcomes that are defined as of high quality through the management lens (irrespective of peer acclaim) rarely succeed. In other words, I have seen that success through promotion in recent times requires the applicant to use a managerial view of the world to frame their accomplishments, again reflecting the masculinist requirement that success needs one to perform "like a man" (ibid). For example, I have seen high performing women colleagues whose individual work has a huge, demonstrable, positive impact on the lives of others, refused promotion to professorial level because of a lack of category 1 research grants. In contrast, I have seen male academics receive promotion based on membership of a large team that secured a category 1 research grant. Once membership of the elite group (management) is established, members accrue cultural, professional and economic capital that further reinforces their distance from workers, creating a "class war from above" [15] (p. 504) that is self-sustaining.

## 4. Individualism

Women academics are operating in an environment which defines career advancement through a neoliberal lens that privileges masculinist behaviours. This shaping of the definition of success is mostly invisible (except to those who challenge it) but nonetheless

extremely powerful. Individualism, another neoliberal trope, also contributes to the invisibility of this masculinist privilege. The workforce is positioned as "race-less, gender-less and ageless and free to choose regardless of their material conditions" [16] (p. 177). Each person is positioned as completely responsible for the choices they make so that gender imbalances in the workforce, so clearly demonstrated in academia by Sims [2] (p. 81), are seen as the consequence of choices individual women make, rather than as the outcome of systemic disadvantage. Women are claimed to be free to choose to pursue a career and potentially remain childless or they can choose to have children and a family. The reality is that either choice leads to negative judgements being made. Women such as the Australian ex-Prime Minister Julia Gillard are publicly castigated for being "wooden" and lacking in empathy because of her choice to remain childless [17]. Women who choose to have children are positioned as less than professional in their behaviour and not able to demonstrate the appropriate level of commitment to their employer [18]—levels of commitment clearly demonstrated by men whose behaviour defines them as more appropriately belonging to the elite group. For example, in the Promotions Committees where I operated as a union observer, despite claims that output was to be judged relative to opportunity, I saw a number of women refused promotion because their output did not meet the full-time required standard, despite the case made in applications for a number of years of reduced work fraction.

## 5. Women in Academia

The neoliberal social imaginary thus shapes the academic world in which women academics operate. For those seeking to build a career in this context, the choices are either to choose to operate within the accepted social imaginary or to challenge and attempt to change it. For those choosing the former path there are considerable risks. In a social imaginary that "positions workers as incompetent, untrustworthy and in need of micro-management to perform effectively, it is difficult to maintain a sense of self efficacy or self worth" [2] (p. 153). One has to develop sufficient emotional intelligence to withstand daily (sometimes hourly) attacks on professional identity. For example, I recall numerous times being castigated for not following a procedure that had not been previously been made known to staff. There were times when I felt I had to apologise in advance every time I had to ask what the procedure is now to do something, placing me perpetually in a supplicant position, rather than in a position of competence and agency. Developing supportive networks of colleagues can help, as can well-chosen acts of fake accountability [14] or subversive compliance [19]. Such work is likely to feel like busy-work which often feels as if it is replacing the real work, a phenomenon Graeber [20] calls bullshit work. In this context, promotion into the professoriate seems increasingly depressingly difficult as not only does one's academic achievements need to be presented, but one's willingness (and success) in dealing with bullshit work. For example, professorial position descriptions include elements of leadership within the university as well as in the wider discipline, but internal leadership positions are increasingly identified as managerial, requiring acceptance of the neoliberal managerial script, its ways of seeing the world and ways of operating in that world. Thus, to achieve promotion, one can operate within the system by buying into the neoliberal social imaginary and positioning oneself as having management potential. This requires an understanding of management discourse, a language I call bullshit [2], and multiple demonstrations through one's behaviour of belonging. It is not uncommon for academic colleagues, "promoted" into managerial positions to be quickly assimilated into the neoliberal managerial group identity where they are required to demonstrate their willingness "to abandon values their own education might have instilled in them and adopt those of the government, public service and university hierarchs they now serve" [21] (p. 10). Such abandonment of previously held values, behaviours and relationships leads academic colleagues left behind to feel betrayed [22]. This is often reflected in the phys-icality of office positions. In my recent experiences, those accepting management roles (including the lowest of middle management) were moved into a different building from

where they had been placed as colleagues. This change in location meant that (or provided an excuse for) these colleagues no longer joined staff at morning tea and were no longer visibly present to engage in conversations and relationship building/networking.

The alternative path is also full of risks. Resisting and/or challenging the neoliberal social imaginary requires women academics to firstly identify the exploitative narratives that shape their experiences and understandings. Exploring the language and the metaphors we use to understand our work is one strategy that can be used here [23]. For example, positioning students as customers means that the best measure of our teaching is student satisfaction—a position that many challenge as an ineffective measure of quality [24], and one that is acknowledged to be biased against women academics [25]. This kind of professional reflection is not easy in a context where academics work increasingly long hours, many more than those for which they are paid [26]. It is possible to argue that increases in workload serve to keep academics in their place as they are too busy to fight management dictates. Certainly, from my own experience, incredible busyness functioned to keep people in their own silos, too busy to chat to others to find that they were not alone in their experiences of bullying and exploitation. I recall fighting hard for a woman colleague who was told that taking Special Study Leave (SSP) meant that she had to fulfil her annual teaching load in the one semester, given that she was on SSP in the other. At the same time there were male colleagues who were granted SSP and did not have to meet the same teaching requirements. A year or so later, another woman colleague came to me in great indignation as she was also told she had to complete her annual teaching load in one semester because she had been approved to take SSP. I was somewhat frustrated as I had tried really hard to mobilise support for the first colleague but failed to do so, and as a consequence, was facing the same battle again. It is important to note here that not challenging inequity (whilst perhaps a necessary survival strategy at certain points on one's life) is tantamount to accepting them, so that in not challenging we become part of the problem ourselves [27]. Resistance and challenge are safer in groups rather than as individual acts [28], and this requires us to actively reach out and network with colleagues. Again, it takes time to build relationships of professional trust and the busy (bullshit) work required of academics makes it difficult to find that time.

Challenging what is can be much more satisfying when there are ideas of how things could be. Hope is a powerful motivator which is fuelled by small successes. Collective acts of resistance can be carefully chosen so that rather than acting like Don Quixote, we chose our battles carefully and seek symbolic acts that are likely to have an impact [29]. Here unionism has a clear place: the academic union (and associated women's group) provides a context where we can network with like-minded others and a safety umbrella under which we can operate with strategic acts of defiance. We can also consider the value of carefully chosen complaints [30]. In my experience, complaints are rarely handled well by management and often tend to be dismissed after little or no investigation (it is a rather ineffective system to have management responsible for investigating complaints made against other management). I remember a woman colleague laying a complaint because she was refused one day of annual leave attached to the Easter break (during her teaching trimester) which she needed to support her children at a school activity. At the same time several male colleagues had attended a conference in the week before Easter and had taken the final day as annual leave to enable them to stay over at the out-of-town venue for the Easter break. The manager who heard the complaint appeared genuinely bemused that one should want to take a day's leave in what was the woman's teaching period, arguing that she should take leave in her non-teaching trimester.

However, all academics have the option of accessing outside sources through the Public Interest Disclosures Act 1994 (PID ACT). Whilst these are subject to a triage system, so are unlikely to receive attention unless they identify a major problem, there is a historical analysis undertaken regularly which identifies trends, and a consistent or growing trend is more likely to receive attention. Again, I make the point that we all need to be held accountable for our actions. The "creeping tide of incremental shocks" [31] to which we

are all subject should not make us accustomed to wrong doing nor create the impression that "the public isn't watching, and isn't interested", thus permitting the "new cavalier approach to accountability" (p. 5) to flourish.

Whilst resistance is essential, it does not result in lasting changes. If we are unhappy with what is, then we need to work together towards changing towards a better alternative. In this post-truth world, this is difficult. Post-truth bullshit often makes grandiose claims that sound great on the surface—claims such as a commitment to gender equity and social inclusion. However, the reality underneath is often very different: gender stereotyping and individualism attributing the cause of gender and cultural inequity at the feet of women and people from minority groups rather than the system are clear examples. We have to learn to delve below the fine sounding words and phrases so that we challenge the platitude-as-an-epiphany [32] bullshit. We can ask awkward questions: what do you mean when you say gender equity? What strategies do you have in place to support this? Can you please say that again in clear English? Spicer [33] suggests we can even fight back using our own bullshit to clog up the administrative system.

Ultimately challenging requires a change in the way that we, and others, think about the world, changing the neoliberal ideology that drives much of the western world. This means becoming skilled at political messaging, where the process of values or emphasis framing [34] can be used to shape the messages we deliver and what people hear. This involves identifying the values we want our messages to trigger in listeners and crafting our message carefully; for example, messages that address individual advantage (you will be better off if you support this/act in this way) trigger individualistic thinking making the audience much less susceptible to empathetic responses. If we want to trigger support for measures to redress systemic disadvantage then we have to trigger values around responsibility, compassion and loyalty. It is also important here to demonstrate to listeners that they would not be alone should they chose to act in an empathetic manner—this is called values priming. Thus, as a collective we need to determine the values we believe are important. I suggest values such as compassion, empathy, social justice, honesty, equity, friendship, loyalty and responsibility are good places to start, and in doing so we need to avoid values such as individualism, competition, social recognition, ambition, authority, power, wealth and success in the way in which we operate in our work environment and, I suggest, in the way in which we live our lives. We can collectively create a new narrative about the good life and, through our example of living it, create the space where others can participate. Marx [35] suggests that part of this is creating our own rituals that embody the values by which we live. We might prioritise chatting with colleagues at morning tea as more important than a looming deadline for completing marking because we believe that being physically and emotionally present for colleagues is an important part of the support we provide to each other, for example.

## 6. Conclusions

Stephens [36] (p. 52) claims that "true liberation is not a matter of simply swapping roles—but of challenging the system." Our role is to work together to collectively challenge the inequities and hidden agendas that oppress women and minority groups in higher education. Our work is multi-faceted. We need to raise awareness, engage in acts of targeted resistance, create spaces to discuss alternative narratives and ways of viewing the world and ideas of the way the world could be, all the time making sure that we are physically and emotionally available to each other to "tell the story that lights the path to a better world" [37].

**Funding:** This research received no external funding.

**Institutional Review Board Statement:** Not applicable.

**Informed Consent Statement:** Not applicable.

**Conflicts of Interest:** The author declares no conflict of interest.

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
