# Peer review of "Women Academics in the World of Neoliberal, Managerial Higher Education"

_societies, doi:10.3390/soc11010025_

Round 1

Reviewer 1 Report

Nice piece. I appreciate the useful suggestions and hopeful tone at the end. Note that on the PDF, the abstract differs from the one on the website and a typo appears in the title (spelling of "education"). Maybe the title should include "academics" after women because it's not about all women in all roles in higher education.

In the website's abstract, which is also the first paragraph of the piece, the author promises to use her own experience "to reflect on why this is happening." But the piece isn't really a memoir. I had hoped to read more personal anecdotes rather than just generalizations from the literature, which are familiar to most readers.

"Acts of targeted resistance" reminds me of Deb Meyerson's (2001) "tempered radicals" -- still pertinent after all these years and worth citing.

Author Response

1: corrected spelling mistake in the title

2: changed title to women academics

3: I have added more personal anecdotes 

Reviewer 2 Report

This is an interesting article that challenges three critical concepts in higher education Neoliberalism Managerialism and Individualism, and while the author makes a strong case, his/her position is weakened by the sole focus upon women. The issues identified in terms of career progression within the University system is not the remit of women, it is experienced by male colleagues also. Perhaps it is relevant to a particular individual/country cultural context. If so, the paper must clarify this 'local' context or locate the argument more broadly. Although the author begins by stating that s/he uses their own experiences to reflect on why women are particularly disadvantaged, there is little if any (One example overall) such personal experiences evident in the paper. This weakens the paper and requires attention.  I imagine that many readers will identify with the issues identified in the paper, i.e., bullshit work as a way of ensuring academics (generally) have no time to engage in conversations etc., but some concrete examples would be really powerful and empowering. Overall, I welcome this paper's contribution to 'troubling' the status quo and hope it empowers others, male and female to come forward. 

Author Response

1: The framework upon which I base this article is explained in much more detail in my book, and yes, it is more generic than solely feminist focused. However, this special edition was specifically addressing women in higher education so I took this focus in the paper. I have clarified this in the introduction

2: I have added more personal examples

Reviewer 3 Report

Comments from reviewer

I enjoyed reading this article, it is interesting, challenging, audacious even. While not my first encounter, it was, refreshing to see the word “bullshit” in various combinations used in a theoretical context. The subject matter, neoliberalism and its impact on HE in general and women in particular, is important and relevant.

The author reflects on why the HE sector s/he knew no longer exists, and “has been replaced by a corporate neoliberal entity” (p.1) that disregards people, women in particular, in favour of “masculinist leadership behavour” (p.2). The authors´ intention is said to be that of gaining understanding of the above changes and use it to support others still working in the sector. The author encourages women to take action collectively in order to eliminate the system disadvantage they face. As promised, s/he finishes off by offering advice to women who wish to build a career in this neoliberal context.

Below I address a few points that I deem as worthy of consideration.  

1.

Given the abundance of definitions of neoliberalism the author´s understanding of the concept, “as the overarching ideology that shapes the way that we look at the world” (p. 1) directs the discussion into an unduly narrow channel.

2.

A better contextualization in relation to existing theoretical background and/or empirical research on the topic would strengthen the article. Therefore, while the author sheds light on why the HE sector has changed, it is done within a narrow framework and this undermines the argument somewhat.

3.

At the outset the author proclaimed that s/he will use her/his own experiences to reflect on why the above is happening. I thus expected to see a personal account where the author would present a personal narrative, a first person approach with concrete examples of how neoliberalism impacted upon him/her in academia. It would have been of interest to see the author apply theoretical frameworks to explore or analyse particular incidents (critical incidents even) relating to her/his experience of working in her “last years” in neoliberal academia”. Another alternative would have been to give concrete examples of the impact of these changes on the author.  Instead the approach strikes me as being rather too general which doesn´t carry quite the same weight as a more personal account would have done. This is more significant here because the discussion is not based on empirical data but is, as the author states, a reflection on the reasons for the momentous changes in the HE sector and their impact on its workers.

4

While I concur with the author and her literary sources that the workforce is [often] “positioned as race-less, genderless and ageless” (p. 2) I am not convinced that as a rule, workers in the HE sector are treated “as cattle fodder” (p.1) or that being promoted into professoriate only requires to be “seen as a successful masculinist manager or is (only?) dependent on one’s willingness in dealing with “bullshit work” (busy-work rather than real work). As such these are generalizations that surely apply in some/many situations, yet not all of them. Several other statements are put forward without underpinning or argument. This needs consideration.

Author Response

1: I agree there are many definitions of neoliberalism so I have defined how I am using the concept so the reader is clear where I stand. It is not in the remit of the paper to take a different approach to neoliberalism

2: I have used literature to supplement the experiences I have provided. The framework I used is explained in much more detail in my book and there is not sufficient room in this article to go into that in any great detail. It is not in the boundary of the paper to create a more comprehensive overview of why the sector has changed but rather to examine the consequences of these changes, particularly in terms of their impact on women academics

3: I have added more personal examples

4: I have used additional examples to illustrate the points the reviewer identified